# A method to estimate absolute odorant concentration of olfactory stimuli

**Lucie Conchou[1¤a], Ioan-Cristian Trelea[2], Jérémy Gévar[1], Christelle Monsempès[1], Anne-Claire Peron[2], Michel Renou[1], Isabelle Souchon[2¤b], Philippe Lucas [1]***

1 Institute of Ecology and Environmental Sciences of Paris, INRAE, Sorbonne Université, CNRS, IRD, UPEC, Université de Paris, Versailles, France, 2 Université Paris-Saclay, INRAE, AgroParisTech, UMR SayFood, Palaiseau, France

¤a Current address: Nattaro Labs, Medicon Village, Lund, Sweden
¤b Current address: INRAE, Université d'Avignon, UMR SQPOV, Avignon, France
* philippe.lucas@inrae.fr

## Abstract

The accurate quantification and delivery of odorant concentrations remain a significant challenge. Traditional methods estimate stimulus intensity based on the amount of odorant in the source, but this does not reflect the actual concentration sent due to variable evaporation rates and delivery devices. This leads to inconsistencies in stimulus delivery, complicating cross-laboratory comparisons, threshold evaluations, and the replication of natural olfactory conditions in the lab. To address this, we present a model based on mass transfer theory to predict the concentration of odorants delivered by a simple and versatile odor delivery system commonly used in insect electrophysiological experiments. The present model, built with adaptable compartments, accounts for airflow, source size, and the physicochemical properties of odorants. It helps to better design and use odor delivery systems, especially for stimuli required to mimic natural odor environments. Calibration uses known partition coefficients. The model also considers the dynamic shape of odor stimuli, which affects neuronal responses and must be carefully interpreted, especially when using tools like photoionisation detectors (PID). This approach was applied to study the impact of a plant volatile known to activate pheromone-sensitive neurons, (Z)-3-hexenyl acetate, on pheromone detection in *Agrotis ipsilon* moths. While interference occurs in laboratory conditions at 160 ppb, such concentrations are unlikely in natural settings, suggesting these effects are less relevant ecologically.

## Introduction

Mapping sensory information to neural activity requires delivering measurable and reproducible stimuli. The level of difficulty in controlling stimuli depends on the sensory modality. Olfactory stimuli are much more difficult to control in intensity and time than visual, auditory and tactile stimuli because both parameters depend on the

**Data availability statement:** All data used within this paper can be found at https://doi.org/10.57745/KFJLG4.

**Funding:** This study was supported by Agence Nationale de la Recherche in the form of a grant awarded to MR (ANR15-CE02-010-01 - Odorscape) and by Agence nationale de la Recherche - France2030 in the form of a grant awarded to PL (ANR-20-PCPA-0007 - PheroSensor). The specific roles of these authors are articulated in the 'author contributions' section. The funders had no role in study design, data collection and analysis, decision to publish, or preparation of the manuscript.

**Competing interests:** The authors have declared that no competing interests exist.

physico-chemical properties of odorant molecules, most critically their volatility and their binding affinity for surfaces. The choice of an odor delivery system is always the result of a compromise between conflicting requirements (e.g. easy screening of many odorants vs delivering them with fast dynamics [1]). Common practical requirements of odor delivery systems include:

- Reliability: constancy of the concentration delivered at the time scale relevant for the study (ranging from milliseconds to minutes) to ensure reproducibility.

- Minimal odor adherence: use low adsorption material (e.g. Teflon) and avoid long tubing that increase adsorption and thus disturb stimulus dynamics.

- Versatility: easy and rapid switch between compounds to be tested (especially important for wide screenings) and ability to deliver compounds with different physico-chemical properties (volatility, solubility in the solvent).

- Economy: pieces of the device should be cheap and readily available. Furthermore, some odorants are costly and/or available only as small aliquots.

- Easy handling: small size to be easily incorporated into electrophysiological rigs.

Most of the behaviors crucial for the survival of insects depend on their olfaction. Understanding how they perceive their chemical environment and find the resources it contains requires the ability to reproduce realistic stimuli in the laboratory. Unfortunately, measurements of sensitivity to odors are difficult to interpret when stimuli are expressed as the dose at the source, which is often the case for studies of insect olfaction. Systems based on flow dilution allow a precise control of the concentration delivered and have been used on vertebrates [2–5] but at the cost of contamination that can arise from adsorption of odorants to the walls of the device downstream the odor sources. Such devices have not been widely adopted for routine use in insect chemical ecology. One reason is the much broader dynamic ranges of insect olfactory receptor neurons (ORNs) compared to their vertebrate counterparts [6] which leads to testing wider ranges of concentrations and makes contamination more critical in insects. Different odor delivery systems have been described to work on insects [7–13]. A common feature of these devices is an airflow that passes through an odor source during the stimulation period and which is integrated into a permanent airflow focused on the insect. Such solutions fulfill the requirements of versatility, economy and easy handling. In most cases, the odor source is either deposited on a filter paper introduced into a Pasteur pipette [14–17], or kept in liquid solution introduced in a container of variable size. In a small volume of headspace (e.g. 0.1 mL in a 1-mL vial [18]) the odorant concentration is reconstituted very quickly after an air puff while in a large headspace (e.g. 24.9 L in a 25-L tank [19]) the odorant concentration is less diluted by an air puff.

Except for insect pheromones, which are generally less volatile than volatile plant compounds (VPCs), paraffin and mineral oil have been largely adopted as versatile non-odorous solvents that enable production of long-lasting sources. However, the intensity of stimuli is commonly expressed as the amount of odorant introduced in the

source and not the concentration delivered to the insect because the relationship between these two quantities is difficult to establish. Such a relationship has been described in two cases: (i) for odorants loaded on a filter paper introduced in a Pasteur pipette based on the boiling point and lipophilicity of the odorants [20] and (ii) for odorants in liquid phase within vials where the concentration of short odor puffs can be reliably approximated either from the saturating vapor pressure when the odorant is used pure, provided the equilibrium has been reached, or from the air-solvent partition coefficient of the considered odorant when it is diluted in a solvent [21]. However, when stimuli are long (seconds) or repeated, the final concentration is the result of headspace dilution by the airflow and a flow of the odorant from the source solution to the source headspace, following a non-trivial relationship that depends on the odor delivery device. The lack of tools for reliably estimating the absolute concentrations delivered using such simple devices has undermined comparisons between studies, or between compounds with different volatilities, as well as the ability to reproduce ecologically-relevant stimuli.

We provide here a calibration method for a small, simple and versatile odor delivery system commonly used in insect electrophysiological experiments. We describe a method to estimate the absolute odorant concentration delivered from a source composed of a vial containing a few mL of odorant solution in mineral oil. This method is based on a body of theoretical work that describes odorant release from a source-solution and its transport by an airflow. From this theoretical corpus and our observations, we show how, for one particular odor delivery system design, a relationship established from a small number of volatile compounds can be generalized to other compounds of know air-mineral oil partition coefficient. We illustrate how source dimension and airflow values are expected to influence the final concentration delivered and source lifetime in order to provide guidelines for the design of odor delivery devices.

We illustrate how our results can be useful to insect sensory ecology research by tackling an issue that is unresolved because of the difficulty to estimate absolute concentrations delivered on the antenna. This question is whether VPCs can interfere with sex pheromone detection by male moths in their natural environment. Male moths use sex pheromones to locate mates in their environment. High specificity and sensitivity are required for this function because amounts of pheromone emitted are minute. In laboratory conditions, some VPCs excite pheromone-sensitive ORNs or inhibit their response to the pheromone, resulting in a reduced detectability of the pheromone signal in their presence [22–26]. Whether VPC effects can take place at concentrations such as those insects experience under natural conditions has been questioned [27]. (Z)-3-hexenyl acetate (Z3HA) is one of the VPCs known to interfere with sex pheromone detection in male *Agrotis ipsilon* moths. ORNs tuned to the main component of the sex pheromone respond to high concentrations at the source (0.1% and above) of Z3HA [22,28]. When pheromone stimuli are delivered in a Z3HA background, mixture suppression is observed, (i.e. pheromone responses are reduced). This suppressive effect at the sensory level has led to the hypothesis that odor landscapes dominated by some specific VPCs may impair mate search by male moths. However, in the absence of reliable absolute quantifications of active odorant concentrations delivered on the antenna and comparison to concentrations measured in nature it has remained unclear whether the concentrations necessary to trigger these effects in the laboratory can possibly be met by moth in natural conditions. Here we have used a calibrated odor delivery device to document the effect of known concentrations of Z3HA on sex pheromone detection by male *A. ipsilon* and compared the active concentrations of this VPC to ecologically relevant ones.

## Materials and methods

All data processing, statistical analysis and simulations were performed under R software unless further mentioned.

### Chemicals and solution preparation

We selected 11 VPCs commonly emitted by plants and relevant for studies of plant-insect interactions, and with differing volatilities: isoprene (CAS 78-79-5, Aldrich 59240, purity >99.5%), ethyl acetate (CAS 141-78-6, Sigma-Aldrich 270989, purity 99.8%), pentanal (CAS 110-62-3, Aldrich 11013−2, purity 97%), (Z)-3-hexen-1-ol (CAS 928-96-1, Fluka 53056, purity >98%), (E)-2-hexenal (CAS 6728-26-3, Aldrich 132659, purity 98%), α-pinene (CAS 80-56-8, Aldrich 147524, purity

98%), (Z)-3-hexenyl acetate (Z3HA; CAS 3681-71-8, Sigma-Aldrich W317101, purity >98%), eucalyptol (CAS 470-82-6, Fluka 46090, purity >98%), linalool (CAS 78-70-6, Aldrich L2602, purity 97%), indole (CAS 120-72-9, Aldrich I3408, purity 97%) and β-caryophyllene (CAS 87-44-5, Sigma-Aldrich W225207, purity >80%). All VPCs were dissolved in mineral oil (CAS 8042-47-5, Sigma-Aldrich 330779, density 0.838 g/mL). This selection of VPCs provides a basis for demonstrating the validity and potential of our model of prediction of the odorant concentration delivered by an odor delivery system. Among the 11 selected VPCs, Z3HA was chosen for electrophysiological experiments to evaluate whether, at ecologically relevant concentrations, it can interfere with sex pheromone detection in *A. ipsilon* moths.

## Determination of air-mineral oil partition coefficients

Gas chromatography coupled with flame ionization detection (GC-FID) analyses were performed on an Agilent 6890 GC-FID, equipped with a CombiPal autosampler (CTC Analytics). For all compounds but indole, we used 2 columns, an Agilent HPInnowax (30 m × 0.53 mm × 1 μm) for headspace analyses and a Varian CPWAX 57 CB (50 m × 0.25 mm × 0.2 μm) for liquid analyses. For indole, we used an Agilent DB-1 (30 m × 0.53 mm × 1 μm) for headspace and liquid analyses. The injector was heated at 250°C. The areas of the peaks were determined using ChemStation software (Agilent Technologies, Santa Clara, USA).

**Quantification of odorant headspace concentrations by GC-FID.** Vials (20 mL) were loaded with odorant solutions of desired volumes and concentrations. They were then stored for a minimum of 12 h onto a thermostated rack set at 22°C in order to allow time for odorant concentration in the headspace to equilibrate. Partition coefficients of VPCs between air and mineral oil were determined by injecting 2 mL of the headspace from these vials onto the GC-FID. For each VPC, we chose between two of the partition coefficient determination methods described by [29].

**Phase Ratio Variation method (PRV).** The PRV method was the method of choice because it is calibration-free and because precise knowledge of the VPC concentration is not necessary. However, it only works for relatively high air-solvent partition coefficients. The VPC of interest was dissolved in mineral oil at 0.01% and aliquots ranging between 30 μL and 1 mL were dispatched between 15 headspace vials. Under these conditions, the inverse of the FID peak area obtained after headspace injection is positively correlated to the phase ratio (i.e. ratio between the volume of solution and the volume of headspace). The partition coefficient is given by the slope of the regression line divided by its intercept [30]. We fitted the PRV equation on observed FID peak areas following the non-linear regression method described in [31], which allows computing confidence intervals for the fitted parameters that are symmetrical on a logarithmic scale.

**Liquid Calibration method (LC).** We used this method whenever the PRV method did not work (VPCs with partition coefficient below $10^{-4}$). First, the FID detector was calibrated by injecting 1 μL of VPC dilutions in ethanol (3 to 5 concentrations distributed at order of magnitude intervals, 3 replicates each). The calibration curve was estimated by linear regression on log-transformed data. In order to measure the partition coefficient, a series of vials received 1 mL each of solutions varying in VPC concentration (3 to 4 concentrations per VPC, 3 replicates per concentration). FID peak area obtained after injection of the vials' headspace was converted into headspace concentration using the calibration curve, and the partition coefficient was computed as the slope of the relationship between headspace and mineral oil concentrations. This slope was estimated by linear regression with zero intercept. For some compounds, we had two independent series of measurements. In that case, we estimated the partition coefficient from both series together, with series introduced as a random factor, using a generalized estimating equations model (function geeglm() of package geepack in R).

## Experimental stimulation system

To develop a calibration method of odor delivery, we selected a simple and versatile device (Fig 1A). The airflow running into the odor-delivery device was regulated to 2.5 bar with a pressure regulator (Numatics 34203065, Michaud Chailly,

 

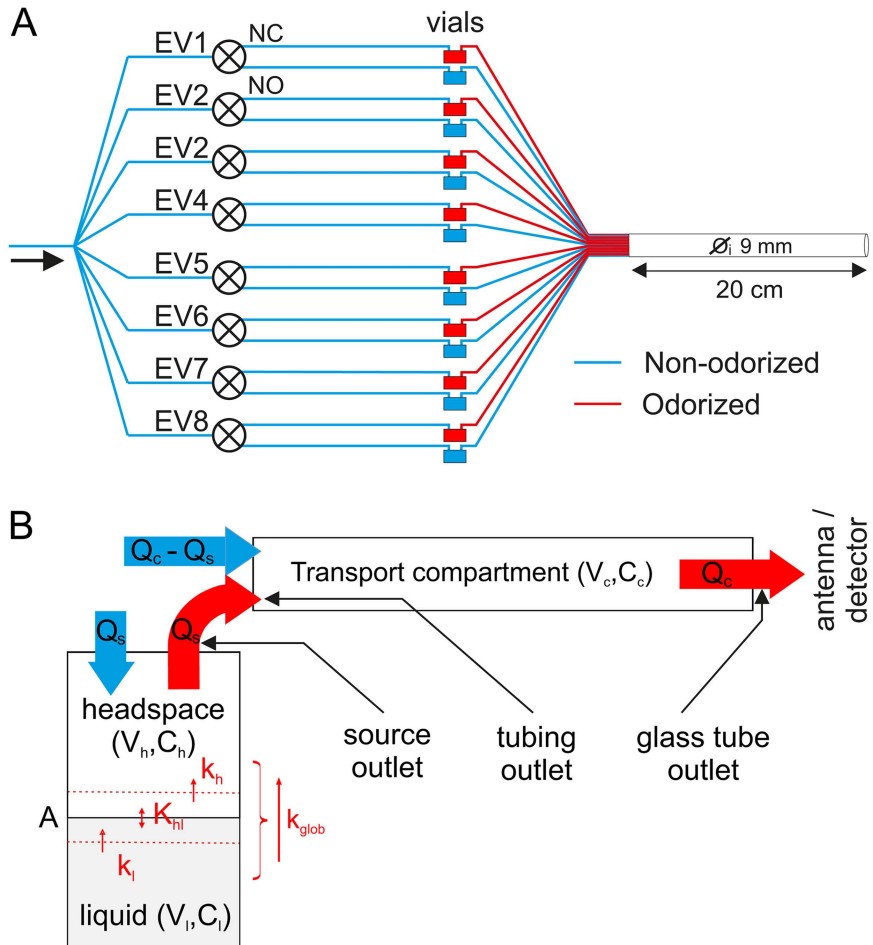

**Fig 1. The odor delivery device. (A)** Diagram of the device. Air is distributed between 8 electrovalves (EV), each of which directs it either through an empty vial (via normally open exit, NO) or an odorized vial (via normally closed exit, NC). All vial outlet tubes are collectively connected to a glass tube that transports the stimuli to the preparation. **(B)** Diagram of the mass transfer model representing odorant transport from one vial to the antenna. The odorant is diluted at a concentration $C_l$ in a volume $V_l$ of mineral oil. The odorant diffuses through the air-solvent interface of area A with a mass transfer coefficient $K_{glob}$ that depends on the air-solvent partition coefficient $K_{hl}$. $k_l$ and $k_h$ are the mass transfer coefficients from bulk liquid to interface and from interface to bulk headspace, respectively. With the headspace of volume $V_h$, the odor concentration has concentration $C_h$. During stimulation, an airflow $Q_s$ passes through the odor source, transports odor molecules, and merges with a non-odorized airflow $Q_c$-$Q_s$ within the glass tube (transport compartment). Between stimuli, no airflow enters the odor source. The delivered airflow $Q_c$ is constant throughout the experiment. $V_c$ is the volume of the glass tube and $C_c$ is the odor concentration at the outlet of this compartment. The 3 black arrows in (B) indicate the locations where odorant concentration were measured with a PID.

Voisins-le-Bretonneux, France) coupled to a 25 µm filter (Numatics 34203065). The incoming air was charcoal filtered (hydrocarbon trap, Cat. #22013, Restek, Lisses, France) and divided by an airflow divider (LFMX0510528B, The Lee Company, Westbrook, CT, USA) into 8 parallel airflows (200 mL/min each), each of which connected to a 3-way electrovalve. The normally closed (NC) exit of each valve was connected to the odor source, a 4-mL vial with inner diameter of 13.2 mm containing 1 mL of odor solution in mineral oil, and the normally opened (NO) exit was connected to a 4-mL empty vial. The odor delivery system was used at 20–22°C. The air inlet and outlet of the vials consist of hypodermic syringe needles (18 G, 1.2×25 mm) inserted through a septum in the lid and a polypropylene Luer connector. All VPC-filled and empty vial outlets are connected to 49 cm-long Teflon tubes (inner diameter 1.23 mm), the downstream end of

which are glued together and connected to a glass tube (200 mm long, 9 mm inner diameter). The airflow delivered was thus constant at all times. Air was humidified for electrophysiological recordings but not for photoionization detector (PID) measurements as it can damage it.

## Modeling of odorant mass transfer within the stimulation system

The odor delivery system described in Fig 1A can be modelled by a succession of compartments, as depicted in Fig 1B. The odor source is represented by two adjacent compartments: source liquid and source headspace, each characterized by their volume ($V_l$ and $V_h$) and separated by an interface, the surface of the liquid, of area A. The airflow $Q_s$, which passes within the odor vial, carries the odorized air to the glass tube, a transport compartment characterized by its volume ($V_c$) and the airflow running through it ($Q_c$).

Mass transfer refers to the movement of chemical species from one location to another, in particular across interfaces [32]. The concepts of mass transfer can be applied to the description of odorant compound release from a liquid matrix (e.g. [33]). In this contribution, we build on this corpus in order to describe odorant release from the source and transport towards the antenna from an odor delivery device.

**Initial distribution of odorants within a closed source: the thermodynamic equilibrium.**  Once the vial is closed after the introduction of the odorant diluted in mineral oil, the odorant volatilizes in the headspace until a thermodynamic equilibrium is established between the concentration in the solvent and the concentration in the headspace (net flux across the interface drops to zero). The laws governing this equilibrium differ depending on the range of odorant concentrations considered [34]. In this paper, we only consider the case of infinite dilution solutions, i.e. concentration so low that solute molecules interact only with solvent molecules. Under such conditions, Henry's law applies and initial odorant concentrations in liquid and headspace compartments (respectively $C_l^0$ and $C_h^0$, mol.m$^{-3}$) are proportional to one another:

$$C_h^0 = K_{hl} * C_l^0$$

(1)

where $K_{hl}$ is the air-solvent partition coefficient, a dimensionless constant. All parameters and variables of the model are listed in Table 1. Air-solvent partition coefficients can be expressed in a variety of ways, and the expression we use here corresponds to the so-called dimensionless Henry volatility, after the terminology by [35]. Partition coefficients depend on the identity of the VPC, the solvent and the temperature.

In practice, the initial odorant concentration inside the headspace of a closed, equilibrated source can be calculated from the concentration of the odorant solution introduced, its volume, and the partition coefficient of the considered odorant between air and the solvent.

For the rest of the manuscript, $K_{hl}$ refers to air-mineral oil partition coefficients.

**Airflow-induced mass transfer inside a source.**  An odorant stimulus is usually obtained by means of running an airflow through the source headspace. The airflow dilutes the headspace and breaks the thermodynamic equilibrium between liquid and headspace concentrations. This results in a net mass transfer of odorant molecules from the liquid to the headspace compartment. At any time, the magnitude of this mass flux, noted $J_{l \to h}$ (mol.m$^{-2}$.s$^{-1}$) depends on the amplitude of the deviation from the thermodynamic equilibrium and on a mass transfer coefficient $K_{glob}$ (m.s$^{-1}$):

$$J_{l \to h} = (K_{hl} * C_l - C_h) * k_{glob}$$

(2)

where $C_l$ and $C_h$ represent liquid and headspace concentration (mol.m$^{-3}$) at the considered time point, respectively. A detailed description of the model leading to equations (2) and (3) is given in Supplementary material.

Global mass transfer can be decomposed into mass transfer from bulk liquid to the interface, and mass transfer from the interface to bulk headspace. These steps are associated to a liquid and a headspace transfer coefficient, $k_l$ and $k_h$

**Table 1. List of variables and parameters of the model.**

| | |
|---|---|
| $A$ | Area between source liquid and source headspace (m²) |
| $C_c$ | Odorant concentrations in the transfer compartment, glass tube (mol.m⁻³) |
| $C_h$ | Odorant concentrations in the headspace (mol.m⁻³) |
| $C_l$ | Odorant concentrations in the liquid, mineral oil (mol.m⁻³) |
| $C_{c,psr}$ | $C_c$ at the pseudo-stationary regime (mol.m⁻³) |
| $C_{h,psr}$ | $C_h$ at the pseudo-stationary regime (mol.m⁻³) |
| $C_h^0$ | Initial odorant concentrations in headspace (mol.m⁻³) |
| $C_l^0$ | Initial odorant concentrations in liquid (mol.m⁻³) |
| $d$ | Diameter of the odor source (m) |
| $D$ | Diffusion coefficient (m².s⁻¹) |
| $J_{l\to h}$ | Mass flux from solvent to headspace (mol.m⁻².s⁻¹) |
| $k_{glob}$ | mass transfer coefficient (M.s⁻¹) |
| $K_{hl}$ | Air-solvent partition coefficient (dimensionless constant) |
| $k_h$ | Mass transfer from interface to bulk headspace (m.s⁻¹) |
| $k_l$ | Mass transfer from bulk liquid to the interface (m.s⁻¹) |
| $M$ | Air viscosity (kg.m⁻¹.s) |
| $P$ | Air density (kg.m⁻³) |
| $Q_s$ | Airflow running through the odor vial during stimuli (m³.s⁻¹) |
| $Q_c$ | Constant airflow running through the transport compartment (m³.s⁻¹) |
| $Re$ | Reynolds number |
| $Sc$ | Schmitt number |
| $Sh$ | Sherwood number |
| $V_c$ | Volume of the glass tube (m³) |
| $V_h$ | Volume of source headspace (m³) |
| $V_l$ | Volume of source liquid (m³) |
| $Y_c$ | Dimensionless odor concentration in the transport compartment |
| $Y_h$ | Dimensionless odor concentration in headspace |
| $Y_l$ | Dimensionless odor concentration in liquid |
| $Y_{h,psr}$ | Dimensionless headspace concentration at dynamic pseudo stationary regime |
| $Y_{h,psr,obs}$ | Dimensionless concentration at source outlet at pseudo-stationary regime |

, respectively. Transfer through the interface itself is considered instantaneous, and the concentrations inside bulk liquid and bulk headspace compartments (i.e. at the exclusion of a thin layer near the interface) are considered uniform at any time. Under these hypotheses, $K_{glob}$ can be written as:

$$\frac{1}{K_{glob}} = \frac{1}{k_h} + \frac{K_{hl}}{k_l}$$

(3)

**Dynamics of odorant concentration in each compartment.** From equation (2), we can deduce a set of differential equations describing how odorant concentrations evolve over time in each of the stimulator's compartments. These are detailed in Supplementary material. Here follows a qualitative description.

The odorant concentration in the source's liquid compartment ($C_l$) decreases due to the loss of molecules through the interface, at a rate that depends on the ratio between the area of the interface ($A$, m²) and the volume of the liquid compartment ($V_l$, m³). See equation (A9) in Supplementary material in S1 File.

On the contrary, changes in odorant concentration in the source's headspace ($C_h$) result from mass gain through the interface and mass loss due to headspace dilution by the airflow running through the source, $Q_s$ (m³.s⁻¹). Mass gain through the interface is again proportional to the ratio between the area of the interface (A) and the volume of the headspace compartment ($V_h$, m³). See equation (A10) in Supplementary material in S1 File.

Finally, odorant concentration inside the transport compartment leading molecules to the antenna of the insect under study, $C_c$, results from mass gain from source headspace and mass loss due to dilution by the carrier airflow ($Q_c$, m³.s⁻¹). See equation (A11) in Supplementary material in S1 File. Odorant concentration delivered on the preparation can be considered as equal to $C_c$ at any time assuming the transport compartment is well mixed due to mixing of 8 air streams at its entry.

For ease of comparison among odorants with very different $K_{hl}$ values, odorant concentrations will from now on be expressed as non-dimensional fractions of $C_l^0$ and $C_h^0$. These are noted $Y_l$, $Y_h$ and $Y_c$ for dimensionless liquid, headspace and transport compartment concentrations, respectively. See equations (A12) to (A15) in Supplementary material in S1 File.

**Simulating the time course of odorant concentration.** Differential equations (8), (9) and (10) which describe the variation of odorant concentration in all 3 compartments over time were fed into the differential equation solver function ode() from package deSolve in R [36], together with the values of the relevant parameters. The initial state was defined as thermodynamic equilibrium inside the source, and zero concentration inside the transport compartment ($Y_l^0 = 1$, $Y_h^0 = 1$, $Y_c^0 = 0$). The output is a table giving the dimensionless concentration at all considered time points inside each compartment.

Varying parameter values allowed to simulate the effect of various odorants ($K_{hl}$) and characteristics of the odor delivery device on delivered concentration time-course (transfer coefficients, source size and airflows).

**Influence of source size and airflow on stimulus concentration dynamics.** In air, odorant mass transfer is predominantly convective. Therefore, the rate of odorant transfer from source interface to source headspace (i.e. the headspace transfer coefficient, $k_h$) is expected to depend on the turbulence level within the source headspace. The level of turbulence inside a fluid is measured by the Reynolds number, which depends on air speed. We calculated Reynolds number inside the source's headspace using the formula for a fluid flowing through a pipe, modified to consider the fact that air speed equals airflow divided by pipe cross-area:

$$Re = \frac{\rho * 4 * Q_s}{\mu * \pi * d} \tag{4}$$

where $\rho$ is air density (kg.m⁻³), $\mu$ is air viscosity (kg.m⁻¹.s), and d is the diameter of the source (m). We used equation (11) to estimate how source dimensions and airflow affect the turbulence level inside the source headspace.

Reynolds number is linked to Schmitt (Sc) and Sherwood (Sh) numbers by a relationship that describes the balance between diffusion and convection in a given system. From this relationship, an equation describing how the headspace transfer coefficient is influenced by the amount of turbulence in the headspace can be deduced:

$$k_h = \frac{D * a * Re^b * Sc^{0.33}}{d} \tag{5}$$

where D is the diffusion coefficient (in m/s, estimated using Wilke-Lee's equation [37]). D depends on compound identity. In a first approximation, we employed the average value for our panel of odorants ($8.54 \times 10^{-6}$ m²/s; ranges between $5.61 \times 10^{-6}$ and $1.16 \times 10^{-5}$). Coefficients *a* and *b* take fixed values that are specific for laminar (1.86 and 0.33), intermediate (0.664 and 0.5) and turbulent regimes (0.04 and 0.75). Sc was calculated from diffusion coefficient and fluid density and viscosity. We again used the average value for the panel of VPCs tested in this work (1.9, ranging between 1.23 and 2.75). *d* is the diameter of the source (m).

Equations (11) and (12) were used to estimate how changes in source dimensions and airflow affect the concentration delivered at the outlet of the source and its time-course.

## Verification of model predictions

We used a 200A MiniPID photo-ionisation detector (Aurora Scientific Inc, Aurora, On, Canada) to measure the odorant concentration delivered by the odor delivery device from sources containing various odorants and concentrations. These measurements were used to verify if the variation in stimulus shape and intensity as a function of odorant $K_{hl}$ value fits the predicted pattern, and to estimate the transfer coefficients for our specific source parameters.

**PID calibration.** For each odorant, a PID calibration curve was established by measuring the PID responses to 4 odorant concentrations spanning 2 to 3 orders of magnitude. To generate odorant concentrations, approximately 100 mL of mineral oil was introduced in a 1-L glass bottle and the exact volume was determined by weighing. The bottle's headspace was rinsed with charcoal-filtered air (1.6 L/min) for a minimum of 5 min. The desired amount of pure odorant was then introduced into the bottle after calculating the amount necessary to reach the desired headspace concentration at thermodynamic equilibrium from the volume of solvent and the $K_{hl}$ value of the considered odorant. The bottles were left to equilibrate overnight at 22°C before use.

First, the PID was left to pump charcoal-filtered air in order to get the baseline value. Then, the bottle containing the calibrated odor concentration was quickly connected to the PID's inlet by means of a hypodermic needle through the bottle's septum and a Teflon tube such that the PID would pump the headspace of the bottle. Another hypodermic needle through the septum allowed charcoal-filtered air to replace the air pumped out by the PID. The PID was connected to the bottle for 1 min (duration inferior to the renewal time of the bottle headspace), and then to a source of charcoal-filtered air. We assumed that the PID's maximum response over the 1-min period corresponds to the response to the initial thermodynamic equilibrium concentration inside the headspace.

**Measurement of delivered concentrations.** Odor sources (2 concentrations per odorant) were prepared and left to equilibrate overnight in the experimental room before they were inserted into the odor delivery system. Each source was used only once. We thus assume that stimuli were delivered after the thermodynamic equilibrium had been reached inside the source. The PID signal was recorded at three locations of the odor delivery device: source outlet, Teflon tubing outlet and glass tube outlet (Fig 1B). At source outlet, the PID's inlet needle was positioned at 1 mm from the outlet Luer. Since the airflow through the source (200 mL/min) is lower than the PID pump's airflow (750 mL/min), charcoal-filtered air was blown on the preparation in order to prevent ambient air from polluting the sample. We took into account the resulting dilution of the odorants entering the PID when calculating concentrations delivered at source outlet: the concentration measured by the PID was multiplied by 750/200. At Teflon tubing outlet, the PID inlet was positioned at 1 mm in front of the tube delivering odorized air from the considered source. At glass tube outlet, the PID was inserted by 1 mm inside the glass tube and centered in its cross section. For the latter two locations, the total airflow from all Teflon tubing or from the glass tube was larger than PID pump's flow, and no dilution of the air delivered occurred.

Once the PID inlet properly positioned, the valve commanding the passage of air through the source was opened for 1 min and the PID signal recorded. The typical shape of the PID signals was a more or less pronounced initial peak, corresponding to the evacuation of the initial headspace at thermodynamic equilibrium, followed by a plateau, i.e., a stabilization of the delivered concentration. For each recording, we calculated the signal intensity at the plateau as the average signal intensity over the last 30 s before closing the electrovalve, minus the average response intensity to charcoal-filtered air during the 30 s before valve opening. The plateau PID responses measured at the three locations were converted into concentration (mol/m³) using the calibration curves established for each VPC. Using equation (7), we converted them into observed dimensionless plateau concentrations.

Fitting of predicted pattern onto observed plateau concentrations at source outlet by combining equations (3) and (14) below, we obtained the equation 11, which was fit onto the observed dimensionless plateau concentrations at source outlet ($Y_{h,psr,obs}$) using nonlinear regression (function nls() in R software):

$$Y_{heq,obs} \sim \frac{A}{A + \exp(B) + \exp(C) * K_{hl}}$$

(6)

where $B = \log(\frac{Q_s}{k_h})$ and $C = \log(\frac{Q_s}{k_l})$

The output was an estimate of $k_h$ and $k_l$ values for our odor delivery device. The use of a log-transformation on the parameters to be estimated allowed us to compute confidence intervals that were symmetrical on a logarithmic scale [31]. We verified the quality of the fit, and therefore the validity of the predicted relationship between odorants' $K_{hl}$ and plateau concentration at source outlet, by calculating the residual sum of squares.

**Proton-transfer-reaction mass spectrometer measures.** To verify if the PID deforms the dynamics of the odorant delivered, we used a proton-transfer-reaction mass spectrometer (PTR-MS; Ionicon Analytik, Innsbruck, Austria). It was operated at E/N 172T. The inlet was heated at 80°C and it drew air at 150 mL/min. The ionization chamber was at 60°C and 405 mbar, with drift voltage 600V, a source tension 4 mA and water flow of 6 mL/min. The acquisition frequency was 20 Hz. Target ions were m/z 99, m/z 81 and m/z 95 for (E)-2-hexenal, linalool and β-caryophyllene, respectively.

## Electrophysiological recordings

**Electroantennography (EAG) recordings.** Moths were immobilized using a pipette tip and dental wax. Glass electrodes were filled with Ringer solution (in mM: glucose 354, KCl 6.4, HEPES 10, $MgCl_2$ 12, $CaCl_2$ 1, NaCl 12, osmotic pressure 450 mOsmol/L, pH 6.5). Recordings were done using a CyberAmp 320 (Molecular Devices, San Jose, CA, USA) and sampled at 1kHz with a Digidata 1440A acquisition board (Molecular Devices) controlled by pCLAMP10 (Molecular Devices). The reference electrode was inserted into the insect's eye, and the recording electrode was set into contact with the tip of one antenna whose last segment had been cut. The biological signal was amplified (×100), low-pass filtered (1000 Hz) and sampled at 1 kHz.

For EAG, we added an electrovalve (EV9) to the described odor delivery device to deliver a known concentration of Z3HA at the plateau with no initial concentration peak (S2A in S2 Fig). This odor delivery device was operated in 3 phases. During phase 1 (equilibration), one upstream electrovalve (EV1 to EV8) was open for 18 s in order to let the concentration reach its plateau at the level of the downstream valve (EV9) which remained closed, so that odorized air was directed to the vacuum vent. During phase 2 (stimulation), the upstream valve remained open and the downstream valve was open for 0.5 s. At the end of this puff, both the upstream and the downstream valve were closed. During phase 3 (rinsing), the upstream valve feeding pure air into to the control source (mineral oil only) was opened, and the downstream valve was opened 5 s later for 500 ms in order to evacuate odorant traces. Each antenna was stimulated with 7 concentrations of Z3HA in increasing order, preceded and followed by a control (mineral oil), leaving 1 min between two successive stimuli. Recordings were done from 10 male and 10 female antennae. The EAG response amplitude to control was subtracted from the response to each Z3HA stimulus and a dose-response was established. For each dose and each sex, we compared the average control-subtracted response to zero using t-tests. Multiple testing was accounted for by adjusting the significance threshold using Benjamini-Hochsberg's procedure to keep the false discovery rate below 0.05. Whenever a significant response to one particular dose was found, we tested whether response intensity differed between sexes using an ANOVA.

**Single sensillum recordings.** *A. ipsilon* males were briefly anesthetized with $CO_2$ and restrained in a Styrofoam holder. One antenna was immobilized with adhesive tape. A glass electrode was inserted into the antenna to serve as a reference. The recording electrode was inserted at the base of a long trichoid sensillum located along an antennal branch where are located the pheromone-sensitive ORNs. Electrodes were filled with (in mM): 172 KCl, 22.5 glucose, 10

HEPES, 3 MgCl2, 1 $CaCl_2$, 25 NaCl, 450 mOsmol/L, pH 6.5 (recording electrode) or: 6.4 KCl, 340 glucose, 10 HEPES, 12 MgCl2, 1 CaCl2, 12 NaCl, 450 mOsmol/L, pH 6.5. Recording hardware and software were identical to those used for EAG recordings. The biological signal was amplified (×1000), low-pass filtered (3 kHz) and sampled at 10 kHz. Spikes were sorted using Spike 2 software (CED, Oxford, Great Britain).

We used the same odor delivery device as for EAG and added an electrovalve, EV10, to deliver pheromone stimuli (0.2 s, 156 mL/min) (S1B in S1 Fig). The sex pheromone, (Z)-7-dodecenyl acetate (Z7-12:Ac), was loaded on a filter paper (10 ng in 1 μL of hexane) introduced in a Pasteur pipette. The tip of the Pasteur pipette was inserted into a hole in the glass tube of the odor delivery device, 5 cm downstream of its proximal end. Z3HA was diluted in mineral oil from $10^{-8}$ to $10^{-2}$ v/v dilution, and pure mineral oil was used a control. EV10 opened 20 s after one of the upstream valves (EV1 to EV8) was open, in order to allow enough time for Z3HA concentration to reach its plateau. Then, a constant background of Z3HA was applied for 20 s. The outlet of the odor delivery system was positioned as close as possible (5 mm) to the recorded sensillum. Z3HA concentrations were tested in increasing order. An interval of 1 min separated two successive background applications. Recordings were carried out on 20 insects and were completed (all Z3HA concentrations tested) for 16 insects.

We calculated the ORN responses to the background by subtracting the spontaneous activity (averaged during the last 5 s before background onset) from the ORN response to background (averaged over the first second after background onset). Pheromone responses were calculated by subtracting the average spiking frequency for 1 s prior to pheromone puff onset) from neuronal activity after the pheromone puff onset (average spiking frequency over 0.5 s). Time window limits were shifted by 0.2 s relative to valve command timings in order to account for the time needed for the odorized air to travel from source to antenna. We tested the effect of stimulus concentration on response to the background and to the pheromone using pairwise paired t tests comparing all possible pairs of background concentrations. Multiple testing was accounted for by adjusting the significance threshold using Benjamini-Hochsberg procedure to keep the false discovery rate below 0.05.

## Results

### Air-mineral oil partition coefficients

We measured the air-mineral oil partition coefficients for our panel of odorants (Table 2). We obtained a large range of volatilities, with $K_{hl}$ values spanning 6 orders of magnitude.

### Model predictions: stimulus shape and dynamic equilibrium concentrations

**Stimulus dynamics are independent from initial concentration.** One step of the modeling process involves expressing the odorant concentration inside each compartment of the odor delivery device as a fraction of the thermodynamic equilibrium concentration inside source headspace, $C_h^0$ (see equations (7) to (10)). The fact that $C_h^0$ does not appear in equations (9) and (10) implies that the time-course of relative odorant concentration inside the source headspace, $Y_h$, and inside the transport compartment, $Y_c$, are independent from the initial odorant concentration in the source headspace. The amount of odorant initially introduced inside the source influences the intensity of the stimulus, but not its shape. The same holds true for the concentration in source liquid compartment (see equations (7) and (8)).

**The dynamic pseudo-stationary regime.** From the initially closed source at thermodynamic equilibrium, passage of air through the headspace triggers a drop in $Y_h$. The time course of $Y_h$ follows an exponential curve, with a sharp decrease followed by a stabilization when dilution by the airflow and mass flow through the interface are balanced (Fig 2C). This dynamic pseudo-stationary regime is maintained as long as the drop in $Y_l$ is negligible ($Y_l \approx 1$). This can be considered true under sufficiently short time scales, and for compounds with a sufficiently small $K_{hl}$ (i.e. when a large fraction of the molecules partition into the liquid, which is validated for all VPCs tested in this work whose $K_{hl} < 10^{-2}$; Table 2). For $Y_l = 1$, we can calculate the headspace concentration at dynamic pseudo-stationary regime, $Y_{h,psr}$ as:

**Table 2. Partition coefficient values measured for the panel of tested VPCs.**

| Compound | Measurement method[1] | Air-Mineral oil partition coefficient ($K_{hl}$) | |
| | | Estimate | 95% confidence interval |
|---|---|---|---|
| Isoprene | PRV | $1.14 \times 10^{-2}$ | $1.02\text{-}1.28 \times 10^{-2}$ |
| Ethyl acetate | PRV | $6.98 \times 10^{-3}$ | $6.28\text{-}7.76 \times 10^{-3}$ |
| Pentanal | PRV | $2.54 \times 10^{-3}$ | $2.27\text{-}2.85 \times 10^{-3}$ |
| (Z)-3-hexen-1-ol | PRV | $7.49 \times 10^{-4}$ | $6.50\text{-}6.63 \times 10^{-4}$ |
| (E)-2-hexenal | PRV | $6.25 \times 10^{-4}$ | $5.76\text{-}6.78 \times 10^{-4}$ |
| α-pinene | Liquid calibration | $3.84 \times 10^{-5}$ | $3.42\text{-}4.25 \times 10^{-5}$ |
| (Z)-3-hexenyl acetate (Z3HA) | Liquid calibration | $2.71 \times 10^{-5}$ | $2.70\text{-}2.72 \times 10^{-5}$ |
| Eucalyptol | Liquid calibration | $1.49 \times 10^{-5}$ | $1.34\text{-}1.65 \times 10^{-5}$ |
| Linalool | Liquid calibration | $7.48 \times 10^{-6}$ | $7.32\text{-}7.64 \times 10^{-6}$ |
| Indole | Liquid calibration | $1.88 \times 10^{-6}$ | $1.78\text{-}1.99 \times 10^{-6}$ |
| β-caryophyllene | Liquid calibration | $1.92 \times 10^{-7}$ | $1.88\text{-}1.96 \times 10^{-7}$ |

[1] See Materials and Methods.

$$\frac{dY_h}{dt} = 0 \Leftrightarrow Y_{h,psr} = \frac{A}{A + \frac{Q_s}{k_{glob}}}$$

(7)

$Y_{h,psr}$ is independent from $C_l^0$ and $C_h^0$, i.e. from the amount of odorant initially introduced inside the source. It only depends on the physical characteristics of the source (interface area and airflow through the headspace) and on the mass transfer coefficient $k_{glob}$.

**Stimulus intensity and shape depend on air-solvent partition coefficient.** Equation (3) implies that the mass transfer coefficient ($k_{glob}$), and thus $Y_{h,psr}$, depend on $K_{hl}$ and therefore on odorant identity. Fig 2A, 2B depict the shape of the relationship between odorant identity and dynamic pseudo-stationary concentration, as expected under equations (3) and (11). We assume that $k_h$ and $k_l$ are independent from odorant identity. At low partition coefficient, the term $K_{hl}/k_l$ of equation (3) is negligible. In this situation, $k_{glob}$ is independent from the compound's partition coefficient and is equal to $k_h$ (left part of the curves in Fig 2A). At high partition coefficient values, the term $1/k_h$ of equation (3) becomes negligible and the global transfer coefficient becomes inversely proportional to the partition coefficient (right part of the curves in Fig 2A). The transition point between $K_{hl}$-dependent and $K_{hl}$-independent regimes depends on the ratio between the liquid- and the air-transfer coefficients: the larger $k_l$ relative to $k_h$, the higher the $K_{hl}$ value at which the transition happens. According to equation (14), the relationship between $k_{glob}$ and $Y_{h,psr}$ is monotonic positive. Therefore, for any given odor delivery system (source dimension and airflow value kept constant), the relationship between $Y_{h,psr}$ and $K_{hl}$ follows the same trend as that between $k_{glob}$ and $K_{hl}$ (Fig 2B). The concentration in the headspace stabilizes at a fraction of the initial concentration $C_h^0$, which depends on the compound's $K_{hl}$ (Fig 2C). At source outlet (Fig 2D), this translates into a stimulus with an initial peak of concentration $Y_h = 1$ (evacuation of the volume of air at thermodynamic equilibrium) followed by a plateau of concentration $Y_h = Y_{h,psr}$ (dynamic pseudo-stationary regime), where the relative height of the peak and the plateau (i.e., the shape) depend on $K_{hl}$.

The theory of mass transfer predicts that the concentration of odorant delivered from an initially equilibrated source should eventually reach a dynamic pseudo-stationary regime that will be maintained until the odorant concentration in mineral oil decreases substantially. It further predicts that the value of this dynamic pseudo-stationary concentration, when expressed as a fraction of initial headspace concentration, is related to the compound's air-solvent partition coefficient. Once its parameters established, this relationship allows predicting the concentration delivered from a source loaded with any compound of known $K_{hl}$. The parameters of the relationship between a compound's $K_{hl}$ and

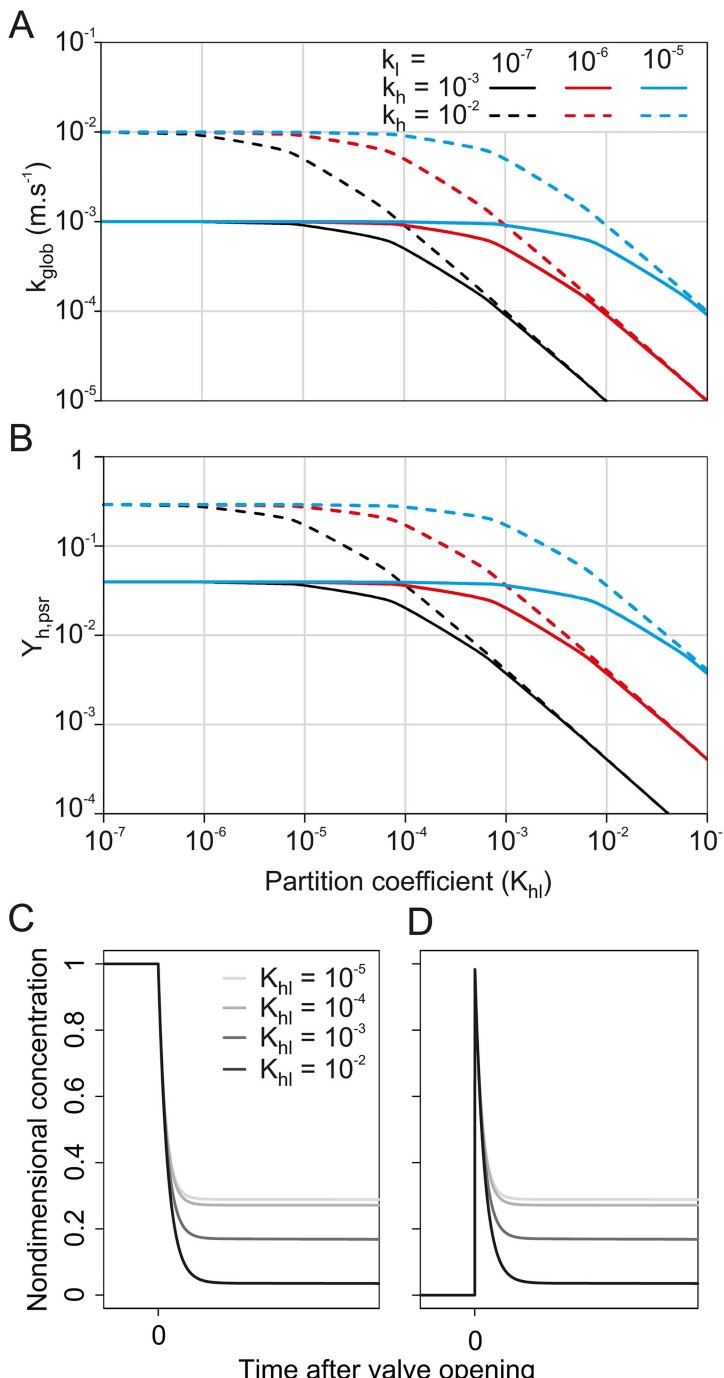

**Fig 2. Results of stimulation system modelling.** Predicted relationship between the partition coefficient ($K_{hl}$) and (**A**) the global mass transfer coeffi-cient from source liquid to source headspace ($k_{glob}$); and (**B**) the dimensionless concentration at dynamic pseudo-stationary regime inside the source's headspace ($Y_{h,psr}$). Different line styles and colors indicate alternative couples of $k_h$ and $k_l$ values. Predicted time-course of the odorant concentration (**C**) inside the source headspace and (**D**) at the source outlet for 4 $K_{hl}$ values. Simulation parameters: in A-D: A = 1.38 cm² and Qs = 200 mL.min⁻¹, i.e. values of the odor delivery device. In C, $V_l$ = 1 mL and $V_h$ = 3 mL, settings of the odor delivery device, $k_l = 10^{-5}$ and $k_h = 10^{-2}$. In D, simulations are for a transport compartment with the same dimensions as a PID inlet needle ($V_c$ = 0.026 mL, $Q_c$ = 750 mL/min) located at source outlet.

$Y_{h,psr}$ include the source's dimension and airflow passing through it, which can easily be measured, as well as the transfer coefficients, $k_h$ and $k_l$, which cannot be measured directly. The air transfer coefficient can be estimated from source size and airflow, as described in the following section. Finally, the model predicts that compounds with high air-solvent partition coefficient will produce stimuli with a more pronounced initial peak than compounds with lower partition coefficients.

### Verifying model predictions

**Concentration delivered at dynamic pseudo-stationary regime.** The values of dimensionless concentrations delivered at pseudo-stationary regime, as measured by a calibrated PID, are summarized in Fig 3. β-caryophyllene and indole were excluded because no stationary regime could be observed after 1 min of stimulation. Equation (13) was successfully fitted on the observed values (non-linear model, residual sum of squares = 0.146), and allowed estimates for $k_h$ and $k_l$ with 95% confidence intervals spanning less than 1 order of magnitude (Table 3). Estimated values of $Y_{h,psr}$ are independent from odorant identity for compounds with $K_{hl} < 10^{-4}$ (Fig 3): the concentration of such compounds delivered at dynamic pseudo-stationary regime represents approximately 30% of the thermodynamic equilibrium concentration inside the source ($Y_{h,psr} = 0.3$). For $K_{hl}$ values above $10^{-4}$, this fraction becomes inversely correlated to $K_{hl}$: $Y_{h,psr}$ reaches 0.04 for isoprene. Comparing $C_{h,psr}$ values calculated using the fitted parameters to the PID measured concentration shows that the concentrations delivered through our odor delivery device can be reliably estimated through this model (residuals less than 1 order of magnitude, Fig 4A). On the contrary, approximating the delivered concentrations from $C_h^0$ alone gives an overestimation of up to 1.5 orders of magnitude.

The concentration delivered on the antenna at the pseudo-stationary regime is expected to be a simple dilution of the concentration delivered by the source. Accordingly, $C_{c,psr}$ calculated as $C_{h,psr}$ divided by 8 (ratio between the air flow at source outlet and the air flow through the glass tube) estimates the concentrations at glass tube outlet with order of magnitude accuracy, while dividing $C_h^0$ by 8 leads to an over estimation (Fig 4B).

**Stimulus shape.** Signal shapes as measured by PID at source outlet for all the test compounds are summarized in Fig 5. The predicted pattern only partially matched the data. The model predicts that the signal should take the shape of an initial peak followed by a plateau (Fig 2D). Furthermore, the relative size of the peak and plateau is directly linked to

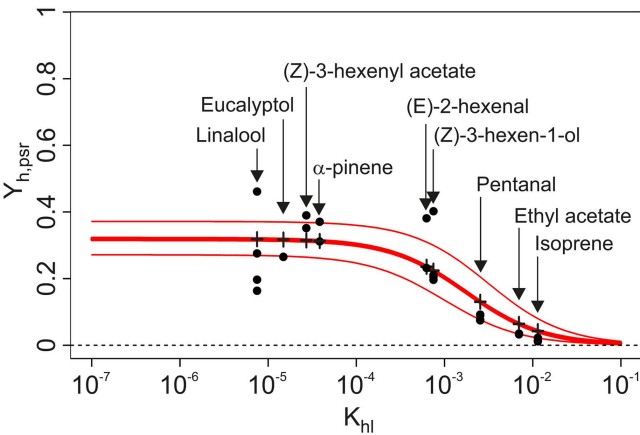

**Fig 3. Dimensionless concentration delivered by odorant sources at dynamic pseudo-stationary regime ($Y_{h,psr}$), as a function of the partition coefficient ($K_{hl}$) of odorants.** Black circles: observed $Y_{h,psr}$ values, measured at source exit using the calibrated PID. Red lines: fit of equation (13) to the observed $Y_{h,psr}$ data (RSS = 0.146, thick: estimate, thin: confidence interval, computed from the limits of 95% confidence intervals for $k_h$ and $k_l$). Black crosses: predicted values for the partition coefficients corresponding to the panel of VPCs (Table 2).

**Table 3. Values of air and liquid transfer coefficients ($k_h$ and $k_l$) estimated from the fit of equation (13) on PID measured values of $Y_{h,psr}$.**

|  | Estimate | 95% confidence interval |
|---|---|---|
| $k_h$ | $1.15 \times 10^{-2}$ | $0.92$–$1.44 \times 10^{-2}$ |
| $k_l$ | $1.24 \times 10^{-5}$ | $0.66$-$2.36 \times 10^{-6}$ |

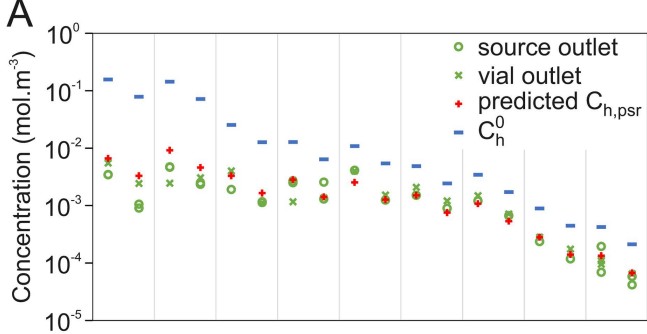

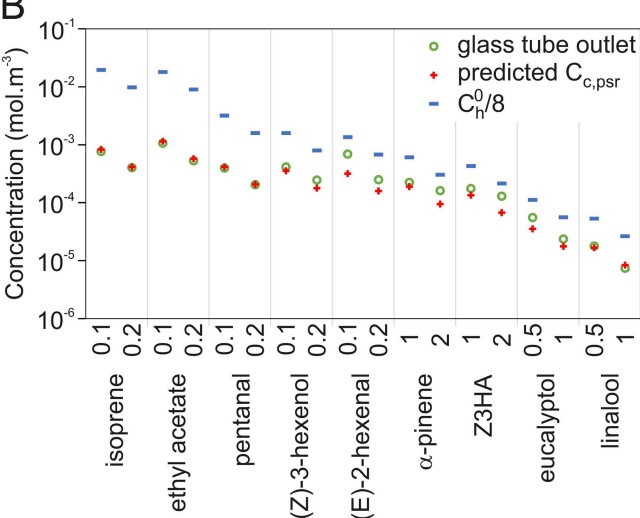

**Fig 4. Comparison of observed (green circles and crosses) and predicted (red crosses) concentration at pseudo-stationary regime. (A)** Comparison of predicted values of pseudo-stationary concentration inside source headspace ($C_{h,psr}$, red crosses) with pseudo-stationary concentrations measured with the calibrated PID at source outlet (green circles) and at Teflon tubing outlet (green crosses). $C_h^0$ is included for comparison. **(B)** Comparison of predicted values of pseudo-stationary concentrations in the glass tube ($C_{c,psr}$, red crosses) with plateau concentrations measured with the calibrated PID at glass tube outlet (green circles). $C_h^0$ divided by 8 (blue bars) is given for comparison.

$k_{glob}$: the higher the global transfer coefficient of VPCs from liquid to headspace, the higher the expected plateau relative to the initial peak. The extreme being, when $Y_{h,psr} = 1$, a simple plateau (peak and plateau of same height). Since $k_{glob}$ varies with $K_{hl}$ (see Fig 2A for predicted pattern), so is the relative size of peak and plateau expected to vary. Fig 3 shows that for our experimental odor delivery device, $Y_{h,psr}$, and therefore $k_{glob}$ are independent from $K_{hl}$ when $K_{hl} < 10^{-4}$, but inversely correlated to $K_{hl}$ when $K_{hl} > 10^{-4}$. Therefore, the prediction is that the relative size of peak and plateau should be constant when $K_{hl} < 10^{-4}$, while the plateau should get smaller relative to the peak as $K_{hl}$ increases, when $K_{hl} > 10^{-4}$.

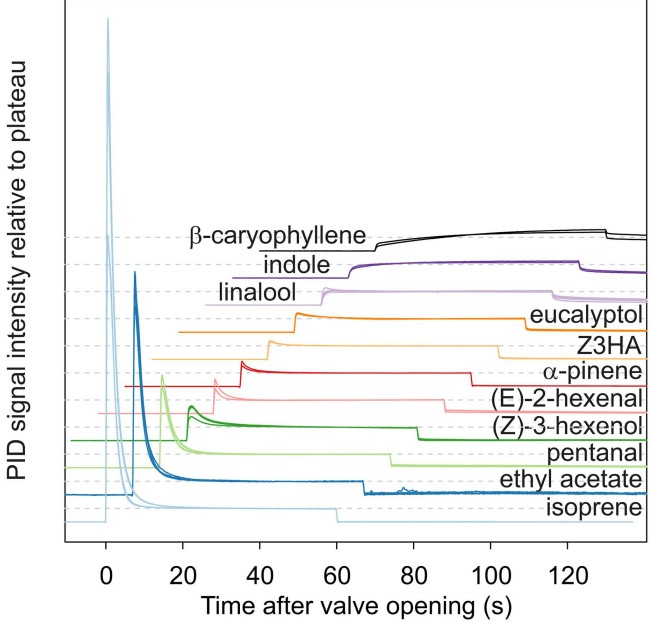

**Fig 5. PID signal recorded at source outlet in response to 60-s puffs of 11 VPCs sorted vertically from top to bottom by increasing air-mineral oil partition coefficient.** Response amplitudes were normalized relative to the plateau. For clarity, from the bottom (isoprene) to the top (β-caryophyllene), each recording is shifted up and to the right.

We do observe curves with an initial peak, with the relative size of the plateau increasing when $K_{hl}$ decreases (Fig 5). However, signal shape continues to change among compounds with $K_{hl} < 10^{-4}$ (α-pinene to β-caryophyllene) instead of stabilizing. In addition, for the least volatile compounds (indole and β-caryophyllene) we observed a third type of shape with no initial peak but a slowly increasing signal that did not stabilize within 60 s of stimulation. Such shapes, characterized by a slow signal increase after valve opening and a slow signal decay after valve closing, are usually attributed to adsorption of volatiles to surfaces of the odor delivery device [1,16,21]. Observing these shapes already at source outlet, in the absence of surface of adsorption between the source and the PID, implies that adsorption occurs either within the source or within the PID. If adsorption occurred primarily within the source, we would expect signal intensity to decay rapidly after valve closing regardless of compound identity, because the airflow passing in the source drops to zero instantly. On the contrary, we observe a slow decay at the source outlet for the least volatile compounds. This slow decay is small but visible for α-pinene to eucalyptol and it becomes substantial for linalool, indole and β-caryophyllene. This observation strongly suggests that physico-chemical interactions between odorants and the PID contribute to the shapes observed in the case of the least volatile odorants. To confirm that the PID deforms the odor signal, we compared the PID responses to those of a PTR-MS. Responses to low-volatility VPCs (linalool and β-caryophyllene) exhibited sharper dynamics at stimulus onset and offset with the PTR-MS than with the PID while no difference of dynamics was observed in the case of (E)-2-hexenal (S3 Fig).

Because the PID low-pass filters the dynamics of the odor puff delivered, we could not compare the observed and predicted concentration time courses.

## Exploring the model for practical recommendations on the design and use of odor delivery devices

Equation (14) shows that A (liquid to headspace interface area) and Qs (source airflow) are involved in determining the value of the pseudo-stationary concentration. $Y_{h,psr}$ tends to 1 when the ratio $Q_s/k_{glob}$ becomes negligible with regards to A,

and decreases when this ratio increases. Assuming the global transfer coefficient ($k_{glob}$) constant, $Y_{h,psr}$ is expected to be higher for sources of a larger diameter, and to decrease when the airflow through the source increases.

Following equations (11) and (12), $k_h$ is expected to increase with source airflow and to decrease when source diameter increases (Fig 6A). Following equation (3), a change in the value of $k_h$ implies a shift in the upper limit of the domain where $k_{glob}$, and therefore $Y_{h,psr}$, are independent from $K_{hl}$. Fig 2B, 2C illustrate how the relationship between $Y_{h,psr}$ and $K_{hl}$ is expected to be influenced by source size and airflow, according to equations (3), (11), (12) and (14), assuming that the liquid transfer coefficient $k_i$ is independent from source size and airflow through the headspace. In conclusion, larger sources and slower airflows lead to higher dynamic pseudo-stationary concentration values that remain independent from $K_{hl}$ over a wider range of $K_{hl}$ values.

**Odorant identity and source lifetime.** One condition to ensure repeatability of odorant stimuli is to replace odorant sources often enough. How long a given source will deliver a constant concentration is therefore a crucial question. Source lifetime can be defined as the amount of time since valve opening after which the delivered concentration drops below 95% of the dynamic pseudo-stationary concentration:

$$t \text{ such as } Y_h(t) = 0.95 * Y_{h,psr} \tag{8}$$

One consequence of the model is that the time course of dimensionless concentrations is independent from initial thermodynamic equilibrium concentrations. Indeed, neither $C_h^0$ nor $C_l^0$ are involved in equations (7), (8) and (10). Therefore, a series of sources prepared with the same odorant at different concentrations are expected to have the same lifetime under this model.

On the other hand, source lifetime depends on the identity of the odorant. Indeed, the smaller $K_{hl}$, the larger the liquid source concentration compared to the headspace source concentration and the longer the lifetime of the source. For the odor delivery device studied here, simulated source lifetimes range from less than a minute for the most volatile compound (isoprene) to several days for the least volatile (β-caryophyllene) (Fig 7). This extreme range of variation implies that the frequency at which sources are replaced must be adapted to the volatility of each individual compound. While

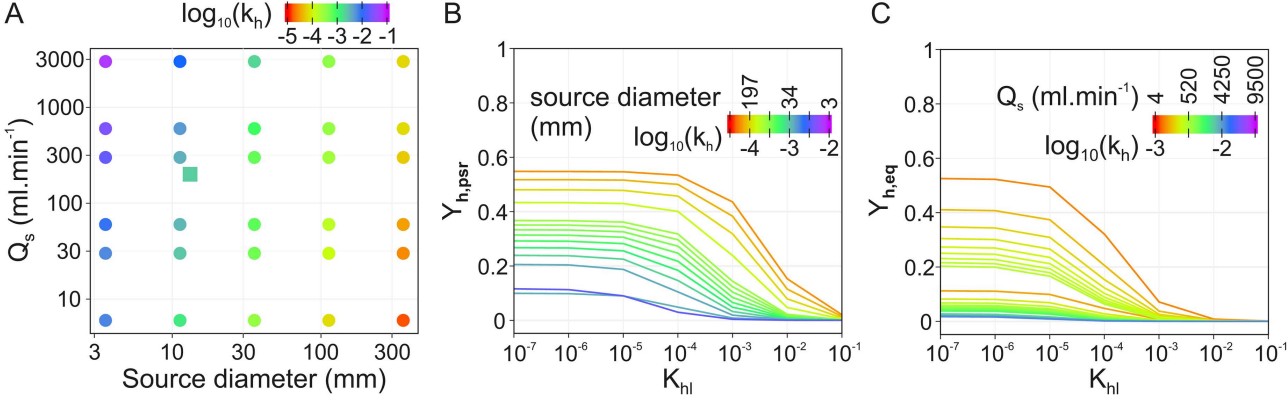

**Fig 6. Expected effect of source diameter and airflow on delivered concentrations. (A)** Theoretical relationship between source diameter (d) and airflow through the odor vial ($Q_s$) and the mass transfer coefficient from interface to headspace ($k_h$), as predicted by equation (12). The point corresponding to the experimental source diameter and flowrate is indicated by a square. **(B)** Effect of source diameter (d) on the expected relationship between $K_{hl}$ and $Y_{h,psr}$, as predicted by equations (11), (12) and (14). Computed for $Q_s = 200\,\text{mL.min}^{-1}$, $k_i = 10^{-7}$, and d varying between 10 and 300 mm (headspace volumes varying between 0.8 mL and 21 L). **(C)** Effect of source airflow $Q_s$ on the expected relationship between $K_{hl}$ and $Y_{h,psr}$, as predicted by equations (11), (12) and (14). Computed for source diameter d = 1.32 cm, $k_i = 10^{-7}$ and $Q_s$ varying between 10 mL min⁻¹ and 5 L min⁻¹. Points and curves are color-coded by $\text{Log}_{10}$ of $k_h$ value.

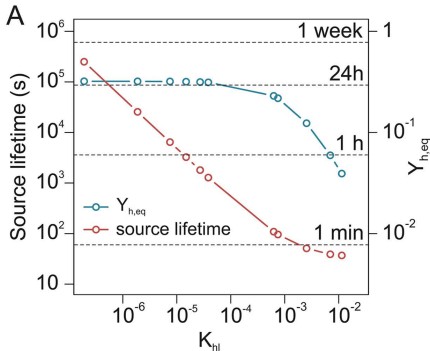 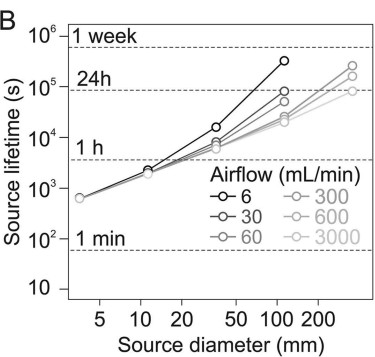

**Fig 7. Simulated evolution of source lifetime. (A)** Relationship between source lifetime and $K_{hl}$ (red). The relationship between $Y_{h,psr}$ and $K_{hl}$ (blue) is given as a reference. Source lifetime is defined as the time t after valve opening at which $Y_h(t) = 0.95 \times Y_{h,psr}$. Source dimension, airflow and transfer coefficients correspond to those of the odor delivery device used in this work. Points correspond to $K_{hl}$ values from to the panel of tested VPCs. **(B)** Source lifetime as a function of source diameter (x axis) and airflow (line shading). Calculated for $K_{hl} = 6.3 \times 10^{-4}$ (value of (E)-2-hexenal), $k_l = 10^{-7}$, and for a source liquid volume = 1/3 of the source headspace volume.

replacement after every day of experiment, or even less, may be safe for the least volatile compounds in the panel, sources loaded with compounds of high volatility must be replaced after every single use.

**How to select source size and airflow?.** According to our model, the concentration delivered at dynamic pseudo-stationary regime depends on source size and airflow, as described by equations (11) and (12). Examining the theoretical trends that can be deduced from these two equations may guide stimulator design. Desirable characteristics and usage of odor delivery device may be as follows:

- $Y_{h,psr}$ close to 1: if this condition is met, $C_h$ can be considered approximately equal to $C_h^0$ at any time. Therefore, $K_{hl}$ becomes the only parameter necessary to estimate $C_h$, and the time course of $Y_h$ becomes approximately flat during source lifetime.

- $Y_{h,psr}$ independent from $K_{hl}$ over the widest possible $K_{hl}$ range. If met, comparisons between compounds of differing volatilities become easier.

- Reasonable source lifetime: replacement of sources at a reasonable frequency. As described above, this issue is more important for compounds of high $K_{hl}$

In the materials and methods section, we describe how values of air transfer coefficient and dynamic pseudo-stationary concentration are expected to vary considerably over a range of reasonable source diameter and airflow values. Maximizing $Y_{h,psr}$ is done either by increasing the source diameter or by decreasing the source airflow (Fig 6B, 6C). The same changes in source dimensions and airflow also maximize the range of $K_{hl}$ values over which $Y_{h,psr}$ is independent from odorant identity.

For a given airflow, source lifetime also increases with source diameter (Fig 7B). The influence of source airflow is very weak for small sources but becomes substantial for larger sources.

**Limits to the applicability of $k_h$ values.** If we apply equations (11) and (12) to the odor delivery system of this study (vial diameter d = 13.2 mm, airflow through the source $Q_s$ = 200 mL/min), we obtain an estimated air transfer coefficient $k_h = 2 \times 10^{-3}$ m/s (see the square on Fig 6A) which is 1 order of magnitude lower than the value found experimentally (Table 3). Equation (12), which we used for estimating Reynolds number, is the formula for fluids flowing in a straight pipe. However, this formula underestimates the level of turbulences for an air stream flowing in the vials through inlets much narrower than the vial itself.

Moreover, the simulations in Fig 6 are made under the assumption that $k_l$ is independent from source dimension and airflow. However, this hypothesis may not always be true: air entering small sources at a high speed may set the oil into motion, increasing convective transport inside the liquid compartment therefore the liquid transfer coefficient $k_l$. Following equation (9), this would modify the effect of source diameter d and airflow $Q_s$ on the threshold between $K_{hl}$-independent and $K_{hl}$-dependent domains in the same direction as their effect on $k_h$. A corollary is that enhancement of $k_l$ value (for example via stirring the solution such as [12]) would expand the range of $K_{hl}$ values for which $Y_{h,psr}$ is independent from $K_{hl}$. Doing so would increase $Y_{h,psr}$ value only for high $K_{hl}$ odorants.

Equations (4) and (5), which allow an estimation of $k_h$, may be useful for designing the dimensions of odor sources. However, they are only reliable for general trends, and direct measurements are required when it comes to calibrating the exact relationship between odorant identity and $Y_{h,psr}$ for a given stimulation system.

### Can interference of plant volatiles with sex pheromone detection by *A. ipsilon* moths occur in the wild?

**Ecologically relevant concentrations of (Z)-3-hexenyl acetate.** Z3HA activates the pheromone ORNs responsible for the detection of the main pheromone component on *A. ipsilon* male antennae [28]. Z3HA is among the most commonly reported compounds from volatile emissions of plant green parts. Therefore, one may expect it to be common in the atmosphere of many ecosystems. To estimate the concentration range that moths are likely to encounter in their natural environment, we made a literature survey of green leaf volatile (GLV) concentrations in the atmosphere of natural ecosystems and agricultural landscapes (S1 Table). To our knowledge, only 3 publications report Z3HA as a confirmed or potential component of the atmosphere. However, its presence may have been overlooked in many other cases because the PTR-MS does not always differentiate between individual GLVs. We identified 16 publications that describe ambient GLV concentrations in various landscapes and their variation over time. The concentrations reported often range between 0.1 and 1 ppbv (parts per billion by volume).

**Antennal detection of (Z)-3-hexenyl acetate.** We measured the EAG responses of male and female adult *A. ipsilon* to calibrated concentrations of Z3HA. The antennae of both males and females showed a significant EAG response only to the two highest tested concentrations (0.1% and 1% corresponding to 160 and 1600 ppbv, respectively; p-values in males $= 2.14 \times 10^{-7}$ and $3.84 \times 10^{-8}$, respectively; in females $p = 5.10 \times 10^{-5}$ and $p = 1.00 \times 10^{-7}$, respectively; Fig 8). At 160 ppbv, males and females responded with the same intensity (p $= 0.58$). At 1600 ppbv, male antennae responded significantly stronger than female antennae (p $= 8.75 \times 10^{-5}$).

**Effect of Z3HA on sex pheromone detection in male A. ipsilon.** We recorded the response of pheromone-sensitive ORNs from male *A. ipsilon* to puffs of sex pheromone presented in a Z3HA background, to evaluate if ecologically relevant concentrations of Z3HA interfere with pheromone detection. ORNs responded to a Z3HA background by an increase in firing frequency reaching its maximum around 1 s after background onset (Fig 9A). The response to background was only significant at 1% (Dunnett's test comparison to control background, p<0.0001, Fig 9B). Pheromone puffs trigger a short and intense increase in firing frequency (Fig 9A). Pheromone responses were significantly reduced under the two highest background concentrations tested (160 and 1600 ppbv) as compared to the control background (Dunnett's test, p $= 0.046$ and p<0.0001, respectively; Fig 9C).

## Discussion

Physiological responses to sensory stimuli depend on the intensity of the stimulus. Olfactory responses are no exception but neurophysiologists working on insect olfaction usually refer to the quantity of the odorant in the source to express the intensity of the stimulus. However, this is a poor estimate of the concentration delivered to the ORNs. Indeed, the evaporation rate of volatiles depends on their physico-chemical properties and can vary dramatically [38–40]. Moreover, for a given compound, the concentration delivered can vary substantially according to the characteristics of the odor delivery device. This hinders i) comparisons of dose-response curves carried out in different laboratories with different devices, ii)

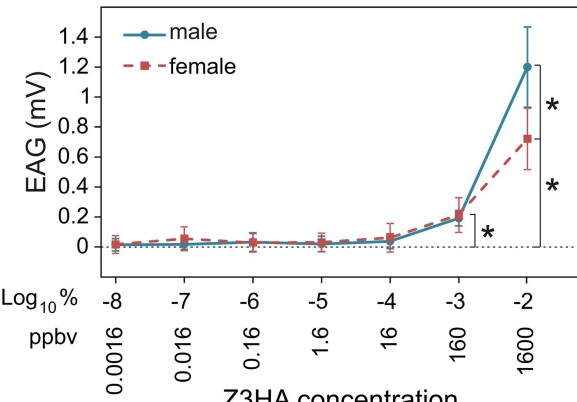

**Fig 8. EAG dose-response of male and female *A. ipsilon* to (Z)-3-hexenyl acetate (Z3HA).** Response intensity to control stimuli was subtracted from the raw response intensities. Stars indicate significant differences to control (one sample t tests) or between males and females (ANOVAs). N = 10 for males and females.

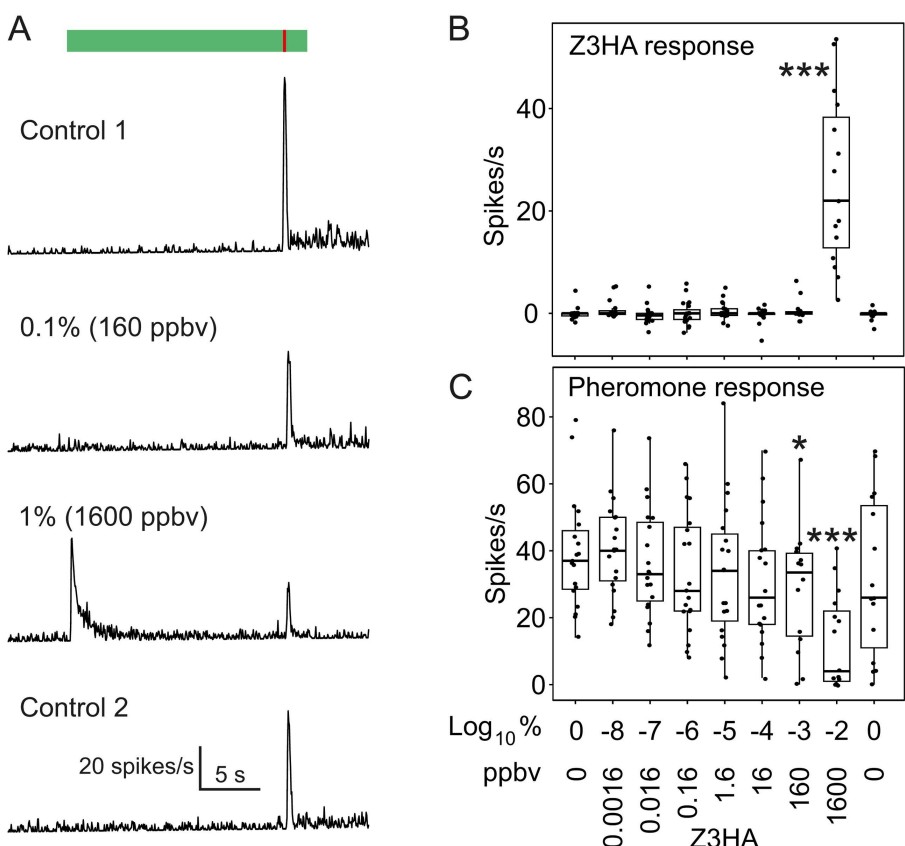

**Fig 9. Response of pheromone-sensitive ORNs from male *A. ipsilon* to pheromone in a background of (Z)-3-hexenyl acetate (Z3HA). (A)** Time course of the firing response to Z3HA background (controls = no Z3HA background were tested before (control 1), and after (control 2) the series of Z3HA backgrounds) and to pheromone puffs (10 ng, 0.2 s). Green box and red line at the bottom indicate the timing of background and pheromone delivery, respectively. Patterns of neuronal activity under the five lowest tested background concentrations are identical to control. **(B)** Responses to Z3HA and **(C)** to pheromone in Z3HA background. Z3HA backgrounds are expressed as both a dilution in mineral oil and a concentration delivered at dynamic pseudo-stationary regime (ppbv). Stars indicate background concentrations under which average firing response differs significantly from that observed under the first control background (Anova followed by Dunnett's test; * = P < 0.05; ** = P < 0.01; *** = P < 0.001). N = 16-20.

evaluation of the response threshold concentration and iii) reproduction in laboratory conditions of olfactory stimulation conditions mimicking natural situations.

As a result, little information is available as to the actual range of concentrations insect olfactory neurons are sensitive to. For short stimuli (<1s), the concentration delivered from a source can be approximated by the thermodynamic equilibrium concentration inside the source, and therefore it can be deduced directly from either the compound's saturating vapor pressure (if the odorant is pure) or its partition coefficient between the solvent used and air [34]. However, short and unique odor stimuli do not satisfactorily reproduce natural odor scenes and some research questions require longer stimuli, either sustained or pulsed. For example, time structure is an essential characteristic of odor signals [41–43], and its proper detection is essential for tracking odor plumes in insects [44–48]. Efforts are thus made to investigate how insect olfactory neurons encode the time structure of olfactory stimuli [16,18,49–53]. Another example is the investigation of how insects detect stimuli of interest over complex odor backgrounds [54–57]. Addressing such questions requires the ability to produce sustained odor stimuli, with stable composition and known concentration over long periods (> 10s). Controlling the absolute concentration of stimuli becomes harder in this context because dilution of the source's headspace by the airflow that runs through it comes into play.

Our work provides a framework for solving this problem using a model based on a simple and versatile device commonly used to deliver odor stimuli by insect olfaction neurobiologists. We show that the theory of mass transfer through an interface predicts that odor concentration inside the source's headspace eventually stabilizes at a predictable fraction of the initial, thermodynamic equilibrium concentration. This fraction depends on the air-solvent partition coefficient of the odorant as well as on the size of the source and airflow passing through it.

The structure of the model as a series of successive compartments, where the change in concentration in each is a function of mass gain from the preceding compartment, dilution by the airflow and mass loss to the following one, makes it easy to adapt the model to any odor delivery systems. Additional compartments can be inserted by adding extra differential equation. This can be useful if one wishes to reproduce the exact geometry of a complex odor delivery system in order to describe with greater precision the time course of odorant delivery on the antenna. This can also allow to model different airflow turbulence levels, by splitting each transport compartment into sub-compartments, and varying their number.

For any given stimulator system and solvent, it is possible to calibrate this relationship using a panel of odorants of known air-solvent partition coefficient, as we demonstrate in the results. Once established, this relationship can be extrapolated to any new odorant, provided its air-solvent partition coefficient is known. Mineral oil and paraffin oil are solvents of choice in olfaction research on plant volatiles because of their olfactory neutrality. However, very little data on air-mineral oil partition coefficients are already available in the literature [40]. There are well-established methods for their measurement [29] but this process is time consuming when repeated for a large number of odorants. For this reason, attempts to extrapolate $K_{hl}$ values from other parameters linked to volatility or affinity to apolar solvents have been made. Cometto-Muniz *et al*. [58] used $K_{hl}$ values estimated from a correlation with saturating vapor pressure. Octanol-water partition coefficient are poorly correlated to $K_{hl}$ [20].

For the sake of guiding the design of odor delivery device, it would be interesting if predicting pseudo-stationary concentration as a function of source size and airflow were possible. This would require to estimate the transfer coefficients ($k_h$ and $k_l$, which represent how fast the odorants are transported through the system), as a function of source size and airflow. The theory of convective and diffusive transport makes such predictions possible (see materials and methods). In particular, we show that larger source headspaces, lower airflows and active stirring lead to pseudo-stationary concentrations that represent a larger fraction of the initial headspace concentration, this fraction being independent from the identity of the compound over a larger range of $K_{hl}$ values. A large source also extends source lifespan. Conversely, low airflow increases the time needed to reach dynamic pseudo-stationary regime, which requires a compromise. Minimizing headspace volume, as in [18], minimizes the time to reach the pseudo stationary regime. However, such predictions may not be accurate enough: for the odor delivery system used here, we observed a discrepancy by 2 orders of magnitude

between predicted and experimental air transfer coefficient values. Therefore, although predictions may be of interest for the sake of identifying broad tendencies, calibration will be necessary every time source size and/or airflow are modified.

Stimulus dynamics is also a hot topic in olfaction research [59]. Indeed, the usual time course as delivered from common odor delivery devices, which is an initial peak followed by a plateau is a problem because it interplays with the neuron's response dynamics, which complicates the comparison of neuron responses to different odorants [16]. A mathematical model of odorant binding on tubing walls can reproduce PID-measured stimulus shapes for a panel of odorants [21]. The PID is the method of choice for documenting odorant stimulus dynamics. The model we describe here allows for predictions of stimulus shapes and how they are expected to vary among volatile compounds. The shapes of odorant signals recorded at source outlet do partially match those predictions, but there are substantial discrepancies with the least volatile odorants. We observed that responses to puffs are more peaked at stimulus onset and offset when recorded with a PTR-MS than with a PID in the case of odorants with low volatility. This demonstrates that for low volatility odorants the observed signal shape are low-pass filtered by the PID, as previously suggested [1,18]. Therefore, although the PID accurately reports the timing of odor puffs owing to its millisecond response time, the shape of PID signals must be interpreted with caution.

For some projects, the perfect shape of olfactory stimuli is a square. More elaborate stimulator designs allow cutting off the initial peak of the stimulus [1,12,18] and generate almost square stimuli. An electrovalve located downstream of the source allows to direct the odorized air first to waste, and then onto to preparation once the dynamic pseudo-stationary regime is reached. Since such devices deliver odorants at dynamic pseudo-stationary concentration, the method we describe here is particularly suited to their calibration.

Once calibrated for odorant concentration, we used the odor delivery device to investigate whether concentrations of Z3HA that interfere with sex pheromone detection in *A. ipsilon* males are likely to be encountered by these moths in their natural environment. A significant EAG response to Z3HA was found in male and females at 160 ppb while in SSR pheromone-sensitive ORNs start responding to this compound at 1600 ppb. This observation suggests that general odorant ORNs are more sensitive to Z3HA than pheromone ORNs. However, Z3HA started interfering with sex pheromone detection by pheromone ORNs from a concentration of 160 ppb. In a literature survey, we found that GLV concentrations reported from actual agroecosystems do not exceed 10 ppb, which suggests on the contrary that 160 ppb is not an ecologically relevant concentration. Even if there may be filaments more concentrated than the maxima reported in the literature, the interactions between the plant-derived signal Z3HA and sex pheromone described in lab conditions at the ORN level are much less likely to occur at natural concentrations. Whether this holds true for all plant-derived volatile compounds and all insects requires to be evaluated.

## Supporting information

**S1 Fig. Schematic representation of the odor source.**
(EPS)

**S2 Fig. Odor delivery devices used for EAG (A) and SSR recordings (B).** Compared to the device presented in Figure 1A and characterized in this work, it included two additional electrovalves (EV9 and EV10) but the length of Teflon tubing was kept similar. For EAG and SSR, EV9 allowed delivering a constant VPC concentration after the elimination of the initial concentration peak. VPC source outlets merge into a low dead volume manifold (MPP8, Warner Instruments, Holliston, MA, USA) connected to EV9. When EV9 was closed, air was directed to vacuum vent and when it was open, air was directed to the side hole of the glass tube via a 1-cm Teflon tubing. Non-odorized vial outlets remained connected to the glass tube and provided the carrier airflow. For SSR, EV10 was used to deliver pheromone pulses on a background of VPC.
(EPS)

**S3 Fig. Comparison of stimulus time courses at the outlet of the source, as observed by a photoionisation detector (PID) and a proton-transfer reaction mass spectrometer (PTR-MS).** The identity and concentration of odorant loaded inside the source is indicated above each panel.
(EPS)

**S1 Table. Literature review of ambient air concentrations of Green Leaf Volatiles (GLVs) in natural/agricultural landscapes.** Publications where authors have tentatively or conclusively identified GLVs in the ambient air of the studied ecosystem, within 35 m above the vegetation canopy. We extracted the estimates of average, median, maximum or range of GLV concentrations. Note that the reported values always represent an average over a certain volume of sampled air. The PTR-MS detects and quantifies the products of a gas sample's ionization, classified by their mass to charge ratio (m/z). When a complex sample such as ambient air is analyzed, it is not possible to discriminate among individual compounds with the same m/z. Identification of atmosphere components by PTR-MS therefore remain tentative unless further analyses confirm which VPCs actually contribute to the detected ions. Ionization of hexenyl acetate produces notably m/z 83 (abundant but unspecific) and m/z 143 (specific but much less abundant [1]). Several other GLVs generate mostly m/z 83 when ionized. This table summarizes the reported concentrations of (A) PTR-MS ions to which Z3HA, (B) PTR-MS ions that are markers of other GLVs, and (C) GLV concentrations estimated by other methods. LOD = limit of detection, sd = standard deviation, ppbv = parts per billion by volume.
(DOCX)

**S1 File. Appendix.**
(DOCX)

## Acknowledgments

We thank Brigitte Pollet and Anne-Claire Peron, the managers of the platform of analytical chemistry of the SayFood Research Unit, for giving access to their equipment and for their very precious help.

## Author contributions

**Conceptualization:** Lucie Conchou, Ioan-Cristian Trelea, Michel Renou, Isabelle Souchon, Philippe Lucas.

**Formal analysis:** Lucie Conchou, Jérémy Gévar, Christelle Monsempès, Anne-Claire Peron.

**Funding acquisition:** Michel Renou, Philippe Lucas.

**Investigation:** Lucie Conchou, Jérémy Gévar, Christelle Monsempès, Anne-Claire Peron.

**Methodology:** Lucie Conchou, Ioan-Cristian Trelea, Philippe Lucas.

**Writing – original draft:** Lucie Conchou.

**Writing – review & editing:** Lucie Conchou, Ioan-Cristian Trelea, Michel Renou, Philippe Lucas.

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
