## [Decision Letter · Decision Letter 0]

11 Aug 2025

Dear Dr. Lucas,

Thank you for submitting your manuscript to PLOS ONE. After careful consideration, we feel that it has merit but does not fully meet PLOS ONE’s publication criteria as it currently stands. Therefore, we invite you to submit a revised version of the manuscript that addresses the points raised during the review process.

**ACADEMIC EDITOR:**

The manuscript needs revision with respect to the reviewers comments and following suggestions.

In manuscript title, specify insect species name with respect to the performed experiments and obtained results.

In methodology section, lacking of information about insect source, collection and maintenance, which used in this study. Please check and add details.

We look forward to receiving your revised manuscript.

Kind regards,

S Ezhil Vendan, Ph.D

Academic Editor

PLOS ONE

Journal Requirements:

3.  Thank you for stating the following financial disclosure: [This work was funded by ANR (grant ANR15-CE02-010-01 "Odorscape")]. 

4. Please include captions for your Supporting Information files at the end of your manuscript, and update any in-text citations to match accordingly. Please see our Supporting Information guidelines for more information: http://journals.plos.org/plosone/s/supporting-information .

Additional Editor Comments:

The manuscript needs revision with respect to the reviewers comments and following suggestions.

In manuscript title, specify insect species name with respect to the performed experiments and obtained results.

In methodology section, lacking of information about insect source, collection and maintenance, which used in this study. Please check and add details.

Reviewers' comments:

Reviewer's Responses to Questions

**Comments to the Author**

1. Is the manuscript technically sound, and do the data support the conclusions?

Reviewer #1: Yes

Reviewer #2: Yes

2. Has the statistical analysis been performed appropriately and rigorously?

Reviewer #1: Yes

Reviewer #2: Yes

3. Have the authors made all data underlying the findings in their manuscript fully available?

Reviewer #1: Yes

Reviewer #2: Yes

4. Is the manuscript presented in an intelligible fashion and written in standard English?

Reviewer #1: Yes

Reviewer #2: Yes

Reviewer #1: As the accurate quantification and delivery of odorant concentrations remain a significant challenge for many chemosensory biologist, the authors have brilliantly developed a model based on mass transfer theory to predict the concentration of odorants delivered by a simple and versatile odor delivery system commonly used in insect electrophysiological experiments by many researchers to estimate the absolute odorant concentration delivered from a source composed of a vial containing a few mL of odorant solution in mineral oil. . Unfortunately, measurements of sensitivity to odors are difficult to interpret when stimuli are expressed as the dose at the source, which is often the case for studies of insect olfaction. But the authors present a model in this study which is helpful to better design and use odor delivery systems, especially for stimuli required to mimic natural odor environments. The model also considers the dynamic shape of odor stimuli, which affects neuronal responses and must be carefully interpreted, especially when using tools like photoionisation detectors (PID). Their method is based on theoretical work that describes odorant release from a source-solution and its transport by an airflow. They are showing how, for one particular odor delivery system design, a relationship established from small subset of volatile compounds can be generalised to other compounds. Authors have illustrated how their results can be useful to insect sensory ecology research by tackling an issue that is unresolved because of the difficulty to estimate absolute concentrations delivered on the antenna. They have used a calibrated odor delivery device to document the effect of known concentrations of Z3HA on sex pheromone detection by male A. ipsilon and compared the active concentrations of this VPC to ecologically relevant ones.

The manuscript is well written in its current form. The authors should take care of the minor suggestions to correct few mistakes:

1. Line 163 -function geeglm() of package geepack in R). Is this an error geeglm() in this line? please correct or specify.

2. Line 265-compartments over time were fed into the differential equation solver function ode() from package. What is ode()?

3. The authors have discussed mineral and paraffin oil in the discussion. What about DMSO that many labs use to dissolve odorants?

4. Use the same nomenclature for (Z)-3-hexen-1-ol everywhere in the manuscript. Check other chemical names as well.

5. Line 837-Table 1 - List of variables and of parameters of the model. Rewrite this as " List of variables and parameters of the model".

6. Line 849- kglob mass transfer coefficient (m.s-1). Make m capital as "M" of mass to keep uniformity with other abbreviations.

Reviewer #2: This study, authored by Conchou et al., investigated the estimation of absolute odour concentration for ORN stimuli in A. ipsilon sp. The introduction and discussion sections of the manuscript are well written. However, the manuscript requires a few modifications to improve its readability.

Major comment:

1. The manuscript is well written by the authors and the English is acceptable. However, the authors need to simplify a few sentences to make them clearer for readers. For example, on line 22, 'the present model or approach'; on line 33: 'The difficulty'?

Line 62: 'Such a solution meets the requirements?'

2. Methods: Line 117: Why did the authors only select 11 VPC compounds for testing, and why did they choose Z3HA for the EAG dose-response and pheromone sensitivity background analysis (Figures 8 and 9)?

3. The authors elaborated on the methodology with a lot of quantification and various formulas for the stimulated system. I suggest that the authors reconsider transferring a few of the more complex formulas to the supplementary information file from the methodology section.

4. The supplementary files are not available for verification (S1 figures).

5. I suggest that the authors provide a conclusion at the end of the discussion rather than with the results.

6. On line 526, the authors state that 'linalool and β-caryophyllene showed sharper dynamics at stimulus onset and offset with the PTR-MS than with the PID'. Why do the authors consider Z3HA rather than linalool and β-caryophyllene?

7. Figure 9: What does ppbv mean in this legend? (C) To the pheromone in the Z3HA background. Are Z3HA backgrounds expressed as both dilution in mineral oil and concentration delivered at a dynamic pseudo-stationary regime (ppbv)?

**Do you want your identity to be public for this peer review?** For information about this choice, including consent withdrawal, please see our Privacy Policy

Reviewer #1: No

Reviewer #2: **Yes: ** Durairaj Rajesh

---

## [Author Response · Author response to Decision Letter 1]

9 Oct 2025

The manuscript is well written in its current form.

We thank the reviewer for this comment and address the following ones point by point below.

The authors should take care of the minor suggestions to correct few mistakes:

1. Line 163 -function geeglm() of package geepack in R). Is this an error geeglm() in this line? please correct or specify.

2. Line 265-compartments over time were fed into the differential equation solver function ode() from package. What is ode()?

geeglm() and ode() are standard functions within the R software environment for data analysis and model computation. The sentences are accurate as stated. We believe that most readers will already be familiar with the specific coding syntax employed by the software R. Providing a detailed explanation of R syntax may go beyond the scope of the manuscript. For these reasons, we propose to leave these sentences unchanged.

3. The authors have discussed mineral and paraffin oil in the discussion. What about DMSO that many labs use to dissolve odorants?

To our knowledge, DMSO has only been used to dissolve odorants in aqueous solutions (e.g. when working with olfactory receptors expressed in Xenopus oocytes or with olfactory receptor neurons in culture). However, the model we describe is applicable to any solvent, provided that the partition coefficients are determined for each solute–solvent pair. We made this point clear by adding "and solvent" in the manuscript (line 671).

4. Use the same nomenclature for (Z)-3-hexen-1-ol everywhere in the manuscript. Check other chemical names as well.

This has been corrected in line 121 of the main text and in Table S1A.

5. Line 837-Table 1 - List of variables and of parameters of the model. Rewrite this as " List of variables and parameters of the model".

This mistake has been corrected.

6. Line 849- kglob mass transfer coefficient (m.s-1). Make m capital as "M" of mass to keep uniformity with other abbreviations.

This mistake has been corrected. 

Reviewer #2: This study, authored by Conchou et al., investigated the estimation of absolute odour concentration for ORN stimuli in A. ipsilon sp. The introduction and discussion sections of the manuscript are well written. However, the manuscript requires a few modifications to improve its readability.

We thank the reviewer for this comment and address the following ones point by point below.

Major comment:

1. The manuscript is well written by the authors and the English is acceptable. However, the authors need to simplify a few sentences to make them clearer for readers. For example, on line 22, 'the present model or approach'; on line 33: 'The difficulty'?

Line 62: 'Such a solution meets the requirements?'

These mistakes have been corrected.

2. Methods: Line 117: Why did the authors only select 11 VPC compounds for testing, and why did they choose Z3HA for the EAG dose-response and pheromone sensitivity background analysis (Figures 8 and 9)?

We clarified it by adding line 126: “This selection of VPCs provides a basis for demonstrating the validity and potential of our model of prediction of the odorant concentration delivered by an odor delivery system. Among the 11 selected VPCs, Z3HA was chosen for electrophysiological experiments to evaluate whether, at ecologically relevant concentrations, it can interfere with sex pheromone detection in A. ipsilon moths.

3. The authors elaborated on the methodology with a lot of quantification and various formulas for the stimulated system. I suggest that the authors reconsider transferring a few of the more complex formulas to the supplementary information file from the methodology section.

We have moved a substantial portion of the formulae to the Supplementary Information.

4. The supplementary files are not available for verification (S1 figures).

There is one single figure in Supplementary Information. The raw data can be downloaded until November 5, 2025 from:

https://filesender.renater.fr/?s=download&token=2ba65324-446d-4f6e-8093-dc5b0b25b6ac

However, please not that they were plotted without any specific treatment.

5. I suggest that the authors provide a conclusion at the end of the discussion rather than with the results.

The conclusion made at the end of the results was moved to the discussion, in the paragraph spanning lines 683-697.

6. On line 526, the authors state that 'linalool and β-caryophyllene showed sharper dynamics at stimulus onset and offset with the PTR-MS than with the PID'. Why do the authors consider Z3HA rather than linalool and β-caryophyllene?

We are sorry, but we do not fully understand the question here.

- Maybe ‘not’ is missing and it should read as: ‘Why don’t the authors consider Z3HA rather than linalool and β-caryophyllene?’.

The choice of molecules tested by PTR-MS was arbitrary; we selected several compounds with different PID dynamics. If the reviewer regrets that we did not select Z3HA for PTR-MS experiments because we used it for electrophysiology, we would like to point out that we took precautions regarding the equilibration times before opening the valve that delivered the stimulus to the preparation.

- If the reviewer did not forget ‘not’, then the answer is that we selected Z3HA because it affects pheromone perception under laboratory conditions.

7. Figure 9: What does ppbv mean in this legend? (C) To the pheromone in the Z3HA background. Are Z3HA backgrounds expressed as both dilution in mineral oil and concentration delivered at a dynamic pseudo-stationary regime (ppbv)?

ppbv stands for parts per billion by volume. We have now defined this abbreviation at its first occurrence (line 612). We thank the reviewer for bringing this oversight to our attention.

---

## [Decision Letter · Decision Letter 1]

6 Nov 2025

A method to estimate absolute odorant concentration of olfactory stimuli

PONE-D-25-31805R1

Dear Dr. Philippe Lucas,

We’re pleased to inform you that your manuscript has been judged scientifically suitable for publication and will be formally accepted for publication once it meets all outstanding technical requirements.

Kind regards,

S Ezhil Vendan, Ph.D

Academic Editor

PLOS ONE

Additional Editor Comments (optional):

Reviewers' comments:

Reviewer's Responses to Questions

**Comments to the Author**

Reviewer #1: All comments have been addressed

2. Is the manuscript technically sound, and do the data support the conclusions?

Reviewer #1: Yes

3. Has the statistical analysis been performed appropriately and rigorously?

Reviewer #1: Yes

4. Have the authors made all data underlying the findings in their manuscript fully available?

Reviewer #1: Yes

5. Is the manuscript presented in an intelligible fashion and written in standard English?

Reviewer #1: Yes

Reviewer #1: (No Response)

**Do you want your identity to be public for this peer review?** For information about this choice, including consent withdrawal, please see our Privacy Policy

Reviewer #1: No

---

## [Editor Report · Acceptance letter]

PONE-D-25-31805R1

PLOS One

Dear Dr. Lucas,

I'm pleased to inform you that your manuscript has been deemed suitable for publication in PLOS One. Congratulations! Your manuscript is now being handed over to our production team.

Kind regards,

on behalf of

Dr. S Ezhil Vendan

Academic Editor

PLOS One